# When to use one-dimensional, two-dimensional, and Shifted Transversal Design pooling in mycotoxin screening

**Xianbin Cheng, Ruben A. Chavez, Matthew J. Stasiewicz** [ORCID] *

Department of Food Science and Human Nutrition, University of Illinois at Urban-Champaign, Urbana, Illinois, United States of America

* mstasie@illinois.edu

## Abstract

While complex sample pooling strategies have been developed for large-scale experiments with robotic liquid handling, many medium-scale experiments like mycotoxin screening by Enzyme-Linked Immunosorbent Assay (ELISA) are still conducted manually in 48- and 96-well plates. At this scale, the opportunity to save on reagent costs is offset by the increased costs of labor, materials, and risk-of-error caused by increasingly complex pooling strategies. This paper compares one-dimensional (1D), two-dimensional (2D), and Shifted Transversal Design (STD) pooling to study whether pooling affects assay accuracy and experimental cost and to provide guidance for when a human experimentalist might benefit from pooling. We approximated mycotoxin contamination in single corn kernels by fitting statistical distributions to experimental data (432 kernels for aflatoxin and 528 kernels for fumonisin) and used experimentally-validated Monte-Carlo simulation (10,000 iterations) to evaluate assay sensitivity, specificity, reagent cost, and pipetting cost. Based on the validated simulation results, assay sensitivity remains 100% for all four pooling strategies while specificity decreases as prevalence level rises. Reagent cost could be reduced by 70% and 80% in 48- and 96-well plates, with 1D and STD pooling being most reagent-saving respectively. Such a reagent-saving effect is only valid when prevalence level is < 21% for 48-well plates and < 13%-21% for 96-well plates. Pipetting cost will rise by 1.3–3.3 fold for 48-well plates and 1.2–4.3 fold for 96-well plates, with 1D pooling by row requiring the least pipetting. Thus, it is advisable to employ pooling when the expected prevalence level is below 21% and when the likely savings of up to 80% on reagent cost outweighs the increased materials and labor costs of up to 4 fold increases in pipetting.

## Introduction

Pooling is defined in this paper as the act of taking an aliquot of equal volume from multiple samples and mixing them. It is intended to rule out a large number of negative samples and detect a few positive samples at a lower cost. Pooling has been widely utilized in various testing scenarios where positive samples are rare, such as high throughput drug screening [1,2],

**Data Availability Statement:** All simulation and data analysis code for this paper, as well as the primary data for validation, can be found at https://github.com/ericxbcheng/Pooling.

**Funding:** This study was supported by the ADM Institute for the Prevention of Postharvest Loss, Postharvest Loss Prevention Graduate Assistantship to RAC (https://postharvestinstitute. illinois.edu/), the Lo Fellowship through the Illinois College of Agriculture, Consumer, and Environmental Sciences to XC (https://fshn.illinois. edu/), and USDA Cooperative State Research, Education, and Extension Service Hatch Project ILLU-698-903 to MJS (https://nifa.usda.gov/ grants). The funders had no role in study design, data collection and analysis, decision to publish, or preparation of the manuscript.

**Competing interests:** The authors have declared that no competing interests exist

detecting human pathogens in clinical samples [3–5], testing for foodborne pathogens in food products [6,7]. To perform pooling, a variety of pooling schemes have been developed including one-dimensional (1D), two-dimensional (2D), N-dimensional, orthogonal pooling, and Shifted Transversal Design (STD) [1,8,9]. Many of these studies on pooling schemes are established on the foundation of group testing problems in combinatorial mathematics and thus are primarily focused on theoretical derivation. Despite advantages of conciseness and generalizability, abstract notations and complex computation in these studies might deter researchers of other disciplines seeking concrete instruction. Moreover, the effect of pooling has been mainly investigated for large-scale experiments (thousands of samples) where sample size is so substantial that it is nearly impossible to assay each individual sample. Medium-scale experiments such as mycotoxin screening in food products, however, are often restricted to hundreds of samples and are usually performed by students and technicians manually pipetting samples into 48- and 96-well plates. On this scale, the reagent-saving benefit of pooling may be counterbalanced by the rising costs of labor, materials and higher risk of human error due to escalating complexity of pooling strategies. Hence, we intend to evaluate the performance and cost-saving effect of pooling in a mycotoxin screening setting and provide experimenters with instruction that leans toward application rather than theory.

This paper employed Monte Carlo simulation technique to study how pooling affects assay sensitivity, specificity, and number of tests and pipetting. To simulate a medium-scale experimental setting, 48- or 96-well assay plates were designated as sample containers as they are frequently used in this setting. As for the simulated experiment, aflatoxin detection in single corn kernels by Enzyme-linked Immunosorbent Assay (ELISA) was selected because aflatoxin contamination in corn, a potential cause of acute toxicosis and liver cancer, is a low-prevalence event in the U.S. [10,11], for which pooling may exhibit its full potential in cost reduction. For countries or regions with higher aflatoxin prevalence, such as Kenya [12], it remains an interesting question how much cost one can save, or more importantly, whether pooling is advantageous at all. Results of pooling simulation for such an experimental setting could be generalized and used to infer the effect of pooling on test performance and cost for other medium-scale experiments. We examined four pooling strategies in this paper: 1D by rows, 1D by columns, 2D, and STD pooling. The first three strategies have been well characterized mathematically for decades and can be readily applied to a wide range of experiments [13]. The last, STD pooling, has been developed more recently and adopted to carry out rapid screening in the field of genomics [14,15]. Our hypothesis is that performing pooling in a medium-scale laboratory setting would lower the cost of experiment while not affecting the sensitivity of the test.

Our results provide guidance on how to select a pooling strategy for 48- and 96-unit experiments based on assay performance and trade-offs between number of test and number of pipetting for a given expected prevalence of positive samples. These results should be useful to experimentalists for evaluating whether pooling is appropriate for their assay.

## Materials and methods

The overall workflow is demonstrated in Fig 1. To start with, mycotoxin levels (aflatoxin and fumonisin) in single corn kernels were measured experimentally. The measured data were then approximated by statistical distributions. Once the fitted distributions were determined to represent the data, they were used to draw random numbers that represented mycotoxin levels. These simulated mycotoxin levels were arranged in various combinations to reflect different plate layouts (48- or 96-well) and prevalence levels (1–47 or 1–95 positive samples). Next, computerized models were constructed to simulate four pooling strategies (1D by row

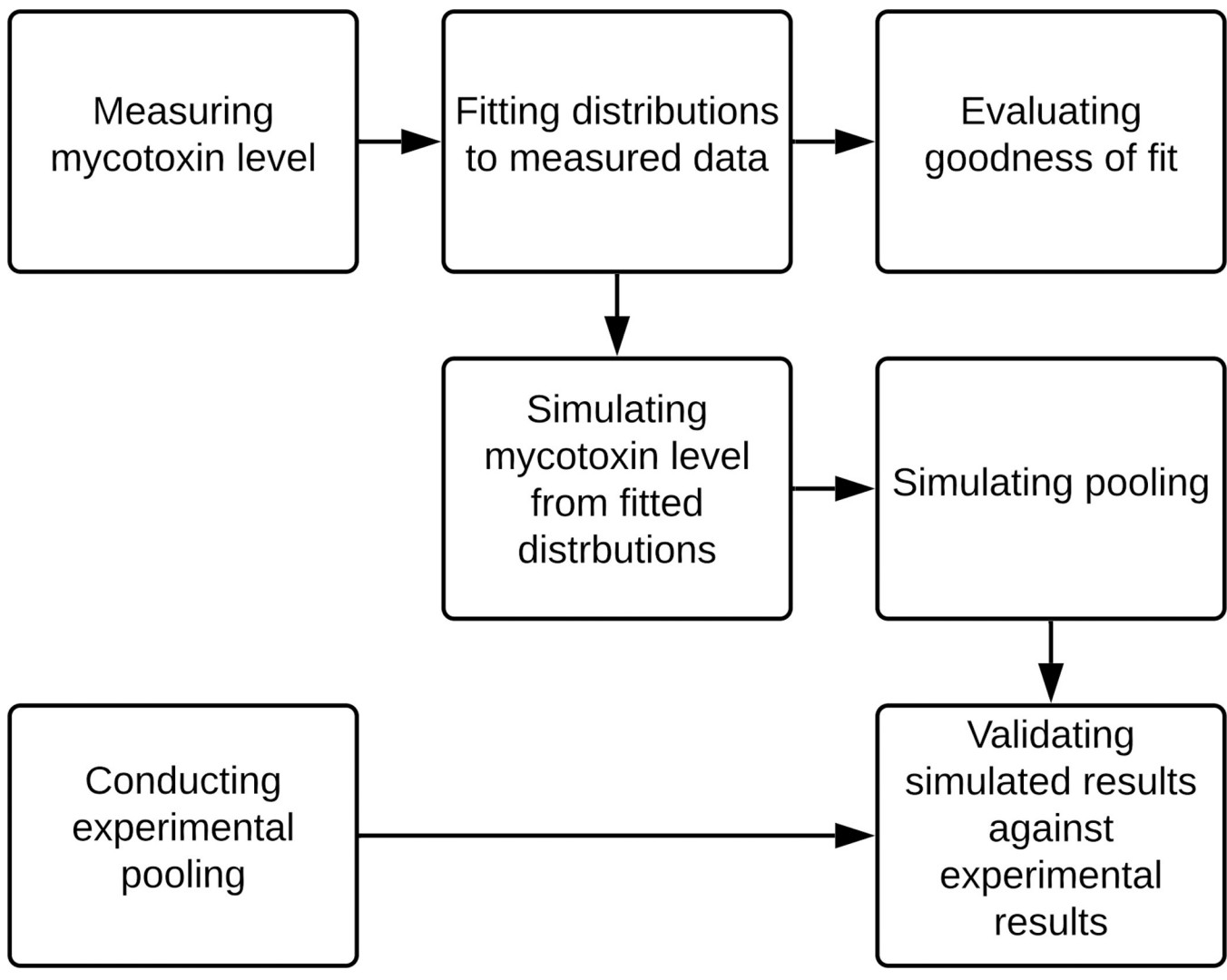

**Fig 1. Flow chart of the overall study design.**

and by column, 2D, STD). With the simulated mycotoxin levels as input, these computerized models ran iteratively and produced the following outputs: sensitivity, specificity, number of tests, and number of pipetting. To validate the outputs from the simulation, experimental pooling was performed on a new batch of corn samples and the experimental results were compared with the simulated results. Details are provided in the following sections.

### Experimental mycotoxin data

Wet chemistry data of Texas commercial corn samples were collected, tested for aflatoxin and fumonisin in single kernels using ELISA kits, and utilized to fit statistical distributions. The corn samples were received in 2017 from the Office of the Texas State Chemist, which they tested in bulk as part of their mycotoxin survey program.

**Aflatoxin.** A total of 432 kernels were randomly selected from three classes of bulk samples with 144 kernels from low-aflatoxin class ($< 20$ ppb), 144 kernels from medium-aflatoxin class (between 20 and 50 ppb), and 144 kernels from high-aflatoxin class ($\geq 50$ ppb). Total

aflatoxins in the kernels were measured by the Total Aflatoxins Quantitative 96-wells ELISA kits (Helica Biosystems Inc., USA)[16]. 1D pooling by row in a 48-well plate was utilized to save on reagent. As a reminder, kernels with < 20 ppb aflatoxin were considered uncontaminated and those with ≥ 20 ppb aflatoxin were classified as contaminated kernels [17]. The limit of detection (LOD) is 1 ppb, which is defined as the expected aflatoxin level in commodity corn that produces assay signal equal to the lowest non-zero concentration standard (0.2 ng/mL). Such an LOD was used throughout the simulation.

After pooling, only 138 out of 432 kernels were individually tested after their pools were identified as positive. Their individual concentration values were used to construct the simulation function for aflatoxin contamination in corn kernels. The remaining 294 kernels were deduced as uncontaminated and excluded from further analysis.

**Fumonisin.** With the same sample selection logic described above, a total of 528 kernels were pooled by row in a 48-well plate and screened for fumonisin by the Fumonisin ELISA Assay (Helica Biosystems Inc., USA). The fumonisin regulatory limit is 1 ppm [18]. After pooling, 93 kernels were individually tested, where 43 kernels were considered contaminated (≥ **1** ppm) by fumonisin and 50 were considered uncontaminated (< **1** ppm). The LOD for this ELISA kit is 0.1 ppm in commodity corn, which corresponds to the 2.5 ng/mL concentration standard. These fumonisin data were used solely for validating the simulated fumonisin distribution.

## Simulation

**Simulation of mycotoxin contamination.** Simulating mycotoxin contamination includes finding an optimal statistical distribution by the maximum likelihood estimation method to fit the experimental data and using that statistical distribution to generate random numbers that represent mycotoxin concentration values.

To fit the aflatoxin data of 138 kernels, aflatoxin concentration in uncontaminated kernels was estimated to follow a modified PERT distribution with the minimum (a) at 0, the mode (b) at 0.7, the maximum (c) at 19.99 and the shape parameter ($\gamma$) at 80. Aflatoxin concentration in contaminated kernels was estimated to follow a customized Gamma distribution with the shape parameter ($\alpha$) at 2 and the scale parameter ($\theta$) at 39980. As a kernel would be considered contaminated only when its aflatoxin concentration was ≥ 20 ppb, the support of this Gamma distribution was shifted by 20 towards the positive direction so that any random number generated from this distribution would be in the range of $[20, +\infty)$.

To simulate fumonisin distribution, a truncated log normal distribution was used to fit the fumonisin data of 93 kernels. Given the positive threshold of 1 ppm, the fumonisin concentration in uncontaminated kernels was estimated to have a mean at -2.75 and a standard deviation at 1.42 with a support on [0, 1]. The fumonisin concentration in contaminated kernels was estimated to have a mean at 3.62 and a standard deviation at 1.74 with a support on $[1, +\infty)$.

Using the distributions above, 48 or 96 random numbers were drawn to represent mycotoxin concentrations. Next, the sequence of mycotoxin concentrations was randomized and reshaped into a matrix to mimic the layout of either a 48-well or 96-well plate.

**Simulation of pooling.** Monte Carlo simulations were implemented to compare the performance and cost among 1D (by rows or columns), 2D, and STD pooling. First, mycotoxin levels were simulated by drawing random numbers from the fitted distributions described above. For simplicity, this study only simulated aflatoxin contamination to evaluate pooling performance and cost. Fumonisin experimental data were only utilized for pooling validation, which will be further described in the validation section. Next, the four pooling strategies were converted into computerized models that could take the simulated aflatoxin level as an input

and return sensitivity and specificity values as outputs. Each of these four models was iterated for 10,000 times over a fully-crossed combination of inputs: 2 plate dimensions (48- and 96-well) and all possible prevalence levels (1–47 or 1–95 positive kernels). Exploring all possible prevalence levels was intended to demonstrate the continuous trend of performance and cost in relation to prevalence and reveal the critical turning point in cost.

**Reproducibility.** Simulations were conducted in R (3.6.1) on Windows 10. In addition to the base R packages, the following packages were also used: tidyverse (1.2.1), MASS (7.3–51.1), EnvStats (2.3.1), and mc2d (0.1–18). The seed for initializing the pseudorandom number generator was set at 123 before running each simulation model.

## Pooling simulation methods

**One-dimensional (1D) pooling.** 1D pooling refers to pooling all the samples of equal volume in a row or a column. In this study, standard 48-well plates (6 rows and 8 columns) or 96-well plates (8 rows and 12 columns) were used as sample layouts. Once the chemical results were obtained for all pools, positive pools were identified by comparing pool concentration to the positive-pool threshold, which was derived as an individual sample's positive threshold divided by pooling size. In the context of aflatoxin detection, an individual sample is considered positive when its aflatoxin concentration reaches 20 ppb [17]. Based on this individual threshold, positive pool thresholds were calculated and presented in Table 1. Any pool whose concentration is higher than or equal to these thresholds is considered a positive pool. Eventually, all individual samples in a positive pool are re-tested to confirm the individual sample status. For any generic pool other than those in Table 1 or for any region with a different mycotoxin regulatory limit, the positive pool threshold can be calculated as $\frac{Positive\ individual\ threshold}{Pooling\ size}$. As a caveat, it is imperative to keep the positive pool threshold larger than or equal to the assay's limit of detection (LOD).

**Two-dimensional (2D) pooling.** 2D pooling is a potential improvement to 1D pooling that may eliminate number of re-tests. First, 1D pooling is performed on both rows and columns. Next, positive pools are identified by the thresholds listed in Table 1. Finally, any individual sample that is present in a positive row-pool and a positive column-pool is considered putatively positive. Re-tests are carried out for these putatively positive samples to confirm their status.

**Shifted Transversal Design (STD).** STD pooling is a complex pooling strategy designed to identify positive samples in a single run [19]. The STD pooling schemes in this study were constructed by following instructions from [14] and [9]. Input and derived parameters that are necessary for constructing and interpreting STD pooling schemes are listed in S2 Table.

**Table 1. Summary of number of pools, pooling size, and positive pool thresholds for one-dimensional (1D)[a] and STD pooling in 48- and 96-well plates.**

| Sample size | Pooling strategy | Number of pools | Pooling size | Positive pool threshold (ppb) |
|---|---|---|---|---|
| 48 | 1D Row | 6 | 8 | 2.50 |
| | 1D Column | 8 | 6 | 3.33 |
| | STD(48; 7; 2) | 14 | 7 | 2.86 |
| 96 | 1D Row | 8 | 12 | 1.67 |
| | 1D Column | 12 | 8 | 2.50 |
| | STD(96; 5; 3) | 15 | 20 | 1.00 |

[a]Two-dimensional (2D) pooling uses the same parameters as 1D row and column pooling.

To reflect performing experiments with 48- or 96-well plates, sample size ($n$) was set as either 48 or 96. To construct a STD pooling scheme for low-prevalence scenarios, the expected maximum number of positive samples ($d$) was set as 1, as it is the lowest possible number. The maximum number of errors expected ($E$) was set as 0. The maximum number of samples allowed to mix in one pool ($m$) was set as 20 so that any positive pool could be detected by ELISA (LOD = 1 ppb) [16].

With the input parameters described above, a list of possible STD pooling schemes was generated, each corresponding to a different number of pools in total. Two more parameters were used to describe the pooling scheme: the number of pools per layer ($q$), and the number of layers ($k$). As the goal of pooling was to economize reagents, the pooling scheme that required the fewest pools was considered optimal and was selected for sequential pooling procedures. When prevalence was at the lowest (only 1 expected positive sample), STD (n = 48; q = 7; k = 2) and STD (n = 96; q = 5; k = 3) would require the fewest pools and were thus chosen for 48- and 96-well plates. Specifically, 14 pools were created for the 48-well plate (pooling size = 7) and 15 pools were created for the 96-well plate (pooling size = 20). A web-based app has been created for generating and visualizing STD pooling scheme based on customized input parameters (see S1 Appendix). Definitions for other STD parameters are listed in S2 Table.

Positive pools are ascertained through the same rule for determining positive pools in 1D and 2D pooling. For aflatoxin detection, the positive-pool threshold for a 48-well plate is $\frac{20}{7} = 2.86$ ppb and that for a 96-well plate is $\frac{20}{20} = 1$ ppb (Table 1).

Eventually, positive individual samples are identified through a 2-step logical elimination procedure. In step 1, if a pool is tested negative, then all its individual samples are marked as negative. For the pools tested positive in step 1, we proceed to step 2 where any individual sample from those positive pools is marked as positive, unless it has already been ruled out as negative in step 1. For example, given a negative pool {A, B, C} and a positive pool {B, C, D}, samples A, B, and C will be determined as negative and D will be positive.

An example of performing STD pooling (n = 48; q = 7; k = 2) with $E = 0$, $m = 20$, d = 1 is illustrated in S1 Fig. Details of constructing this pool are provided in S1 Appendix.

## Data processing and analysis

**Sensitivity and specificity analysis.** In this study, sensitivity (Sn) is the number of contaminated kernels correctly identified as contaminated divided by the total number of contaminated kernels or $Sn = \frac{TP}{TP+FN}$ and specificity (Sp) is the number of healthy kernels correctly identified as healthy divided by the total number of healthy kernels or $Sp = \frac{TN}{TN+FP}$. It is assumed that individual testing has 100% sensitivity and specificity. The detailed definitions for all the abbreviations are listed in Table 2.

**Table 2. Glossary of evaluation metrics.**

| Name | Abbreviation | Definition |
| --- | --- | --- |
| Actual positive | P | The total number of contaminated kernels |
| Actual negative | N | The total number of healthy kernels or $n − P$ |
| True positive | TP | The number of contaminated kernels that are correctly identified |
| True negative | TN | The number of healthy kernels that are correctly identified |
| False positive | FP | The number of healthy kernels misclassified as contaminated |
| False negative | FN | The number of contaminated kernels misclassified as healthy |
| Putatively positive | $TP + FP$ | It can be derived from $Sn \times P + (1−Sp) \times N$ |

**Cost analysis.** Two types of cost were analyzed for each pooling strategy, including the cost of reagent and the cost of pipetting. The experimental procedure could be generalized as three steps: making pools from individual samples, transferring pools into ELISA assay wells, and performing individual sample re-tests for positive pools. The cost of reagent was defined as the number of assays needed to ascertain the status (positive or negative) of all individual samples. The cost of pipetting is the number of liquid transfers by pipetting needed to create the pools, then load the assay wells with the pooled samples and any subsequent re-tests. For the simplicity of the analysis, it was assumed that there was no error in the chemical assay and re-tests would only be carried out for individual samples in positive pools to ascertain the sample status. Calculation of these two costs can be expressed as formulae containing true positives (TP) and false positives (FP) (Table 3). The minimum and maximum costs observed in the simulation were also reported to demonstrate the optimal cost-saving effects and the worst-case scenario.

## Validation

**Validation of simulated mycotoxin data.** Generically, validating simulated mycotoxin data against experimental data (i.e. evaluating goodness of fit) can be summarized in two steps. First, one draws a sequence of numbers from a positive-kernel representing distribution and a negative-kernel representing distribution respectively and concatenates them into one sequence. The ratio of simulated positive and negative kernels should be equivalent to the prevalence level of the experimental data. Second, a Wilcoxon rank sum test with continuity correction is conducted to test whether the simulated distribution is significantly different from the empirical distribution ($\alpha = 0.05$).

Given that the prevalence of aflatoxin contamination in the experimental data was approximately 4%, $9.6 \times 10^5$ random numbers were drawn from the modified-PERT distribution ($a = 0$, $b = 0.7$, $c = 19.99$, $\gamma = 80$) to represent healthy kernels and $4 \times 10^4$ random numbers were drawn from the customized Gamma distribution ($20 + Gamma(\alpha = 2, \theta = 39980)$) to represent contaminated kernels. These two sequences of number comprised a simulated sample of 1 million kernels with 4% prevalence of aflatoxin contamination.

To simulate fumonisin contamination with an empirical prevalence of 46%, 1 million random numbers were drawn where $5.4 \times 10^5$ numbers followed a truncated log normal

**Table 3. Summary of reagent cost (number of assays) and pipetting cost (number of pipetting) of four pooling strategies in 48- and 96-well plates.**

| Plate | Pooling strategy | Reagent cost (number of assays) | | | | Pipetting cost (number of pipetting) | | |
|---|---|---|---|---|---|---|---|---|
| | | Formula[a] | Min[b] (Change) | Max[b] (Change) | Critical prevalence[c] | Formula | Min[b] (Change) | Max[b] (Change) |
| 48 | 1D Column | 8 + TP + FP | 14 (29%) | 56 (117%) | 10 (21%) | 48 + 8 + TP + FP | 62 (129%) | 104 (217%) |
| | 1D Row | 6 + TP + FP | 14 (29%) | 54 (113%) | 11 (23%) | 48 + 6 + TP + FP | 62 (129%) | 102 (213%) |
| | 2D | 6 + 8 + TP + FP | 15 (31%) | 62 (129%) | 10 (21%) | $2 \times 48$ + 6 + 8 + TP + FP | 111 (231%) | 158 (329%) |
| | STD (48; 7; 2) | 14 + TP + FP | 15 (31%) | 62 (129%) | 10 (21%) | $14 \times 7$ + 14 + TP + FP | 113 (235%) | 160 (333%) |
| 96 | 1D Column | 12 + TP + FP | 20 (21%) | 108 (113%) | 20 (21%) | 96 + 12 + TP + FP | 116 (121%) | 204 (213%) |
| | 1D Row | 8 + TP + FP | 20 (21%) | 104 (108%) | 17 (18%) | 96 + 8 + TP + FP | 116 (121%) | 200 (208%) |
| | 2D | 8 + 12 + TP + FP | 21 (22%) | 116 (121%) | 18 (19%) | $2 \times 96$ + 8 + 12 + TP + FP | 213 (222%) | 308 (321%) |
| | STD (96; 5; 3) | 15 + TP + FP | 16 (17%) | 111 (116%) | 12 (13%) | $15 \times 20$ + 15 + TP + FP | 316 (329%) | 411 (428%) |

[a] Each formula is used to calculate a specific cost. TP denotes the number of true positives and FP denotes the number of false positives.

[b] "Min" and "Max" represent the minimum and maximum number of assay/pipetting from the 10,000 iterations. Each change is calculated as the cost divided by the size of assay plate. These *results* are presented in the method section to facilitate comparison between theoretical and empirical values

[c] Critical prevalence is defined as the number (or proportion) of positive kernels that would cost as much reagent to test with pooling as it would cost to test without pooling.

distribution ($\mu = -2.75$, $\sigma = 1.42$, min = 0, max = 0.99) to represent healthy kernels and $4.6 \times 10^5$ numbers followed a truncated log normal distribution ($\mu = 3.62$, $\sigma = 1.74$, min = 1, max = $+\infty$) to represent contaminated kernels.

**Validation of simulated pooling sensitivity and specificity.** Experimental validation was conducted to prove the pooling simulation can accurately estimate the pooling sensitivity and specificity. Due to limited available single-kernel samples and assay reagents, STD pooling was validated with aflatoxin-contaminated samples whereas 1D and 2D pooling were validated with fumonisin-contaminated samples. With the individual and pooling test results combined, the experimental sensitivity and specificity were calculated. These experimental metrics were then compared against the simulated sensitivities and specificities range to determine whether the range of simulated metrics successfully encompassed the experimental ones. More specifically, the simulation was considered valid only when the experimental metrics fall within the inner fence of the simulated range. The inner fence is defined as the range of [$\boldsymbol{Q1} - \boldsymbol{1.5} \times \boldsymbol{IQR}$, $\boldsymbol{Q3} + \boldsymbol{1.5} \times \boldsymbol{IQR}$], where Q1 is the 25$^{th}$ percentile, Q3 is the 75$^{th}$ percentile, and IQR is the interquartile range (Q3 – Q1).

To validate 1D and 2D pooling strategies, 48 corn kernels were individually tested for fumonisins ($B_1$, $B_2$, $B_3$) by the Fumonisin ELISA Assay kit (Helica Biosystems Inc., USA). Next, 6 row pools and 8 column pools were formed and tested, followed by positive pool identification using the rule described above. As a reminder, all the kernels in a positive pool are putatively positive for 1D pooling, and only the kernels at the intersection of positive row pools and column pools are putatively positive for 2D pooling.

To validate the STD pooling strategy, 48 corn kernels were pooled using the STD (n = 48; q = 7; k = 4) pooling scheme. This specific scheme was used because prior experiments suggested a prevalence of 6%, at which this pooling scheme would require the least number of assays. After pooling and determining putatively positive samples, these 48 kernels were individually tested for total aflatoxins following the same protocol described above.

Given the knowledge of true positives from the individual tests, the experimental pooling sensitivity and specificity were calculated through the equations described above. To simulate pooling under the same experimental conditions, the simulation model was supplied with the corresponding input parameters (e.g. $n = 48$, same number of positive kernels as discovered in the experiment, etc.) and was iterated for 10,000 times.

# Results

## Simulated mycotoxin data mimic reality

Experimental mycotoxin data were fitted with optimal statistical distributions, from which simulated data were drawn and compared to the experimental data by the Wilcoxon rank sum test.

Aflatoxin levels in single corn kernels were remarkably skewed. Among the 432 total kernels, 294 kernels were marked as uncontaminated due to negative pooling results and 138 kernels were individually assayed. 6 kernels (4%) had $\geq 20$ ppb aflatoxin and 132 kernels (96%) had $< 20$ ppb aflatoxin (S2 Fig). The median aflatoxin concentration was $4.0 \times 10^4$ ppb for contaminated kernels and 0.74 ppb for uncontaminated kernels.

As for fumonisin, 93 out of 528 kernels were tested individually where 43 kernels (46%) had $\geq 1$ ppm fumonisin and 50 kernels (54%) had $< 1$ ppm fumonisin. The median fumonisin concentration was 36 ppm for contaminated kernels and 0.07 ppm for uncontaminated ones (S3 Fig).

According to the Wilcoxon rank sum test with continuity correction, there was no sufficient evidence to prove the simulated mycotoxin distribution was different from that of the

experimental data (p-value = 0.343 for aflatoxin; 0.768 for fumonisin). Thus, these simulated mycotoxin data mimic observations in reality and are appropriate for pooling simulation.

### Pooling performance

Pooling performance was evaluated by the sensitivity and specificity to detect a positive well in an assay plate with different levels of prevalence (defined as the proportion of kernels with > 20 ppb aflatoxin in an assay plate) [17]. We analyzed pooling for both 48- and 96-well plates and present figures for the 48-well plates in the main body of the text and 96-well plates in Supporting Information. As a reminder, sensitivity is defined as $\frac{TP}{TP+FN}$ and specificity is defined as $\frac{TN}{TN+FP}$.

For the 48-well plate, sensitivity remains 100% for all types of pooling while specificity decreased as the contaminated kernel count increases (Fig 2). Median specificity starts decreasing from 96% for 1D pooling by columns, 91% for 1D pooling by rows, and 100% for both 2D and STD pooling. As the prevalence level increased, specificity for these four pooling strategies dropped to 0%, eventually. While the median specificity for 1D pooling by rows decreased the fastest, reaching 0% when 12 kernels were contaminated, the median specificity for 2D pooling decreased the most slowly, reaching 0% when 18 kernels were contaminated.

In terms of the 96-well scenario, sensitivity also remained 100% and a similar decreasing trend was observed in specificity (S4 Fig). Median specificity started decreasing from 95% for 1D pooling by columns, 91% for 1D pooling by rows, and 100% for both 2D and STD pooling. Unlike the 48-well scenario, median specificity for STD pooling decreased the fastest, reaching 0% when 13 kernels were contaminated, while median specificity for 2D pooling decreased the most slowly, reaching 0% when 30 kernels were contaminated.

### Cost of reagent

Pooling strategies are expected to have the minimum reagent cost at lowest prevalence and have increasing reagent costs as increasing prevalence of positives requires additional re-testing; at some critical point the number of assays required for pooling, which is a proxy for reagent cost, may surpass the number of assays required for un-pooled testing.

In the case of 48-well plates, pooling by any of the four strategies could, at lowest prevalence, reduce the consumption of reagent to around 30% (14–15 assays) compared to what would be needed without pooling (48 assays) (Fig 3 and Table 3). When the number of positive kernels reached a critical prevalence (around 10 positive kernels or 21%), all pooling strategies would require approximately the same number of assays as does non-pooled testing. Beyond the critical prevalence level, pooling strategies no longer saved assays and started to require more assays than testing without pooling. Eventually, the number of assays reaches a plateau of around 55 units for 1D pooling and 62 units for 2D and STD pooling.

For the 96-well plates (S5 Fig and Table 3), pooling could reduce the number of assays to around 20% (16–21 assays) compared to testing without pooling (96 units of reagent). Similar to the case of 48-well plates, as the prevalence level increased, each pooling strategy would reach a critical point where the cost-saving effect by pooling became marginal. While STD pooling reached the critical point the fastest (12 positive kernels or 13%), 1D pooling by columns reached it the slowest (20 positive kernels or 20%).

### Cost of pipetting

The number of pipetting needed for pooling was invariably higher than that for without pooling (Figs 4 and S6). For 48-well plates, 1D pooing would require 1.3–2.2 folds more pipetting

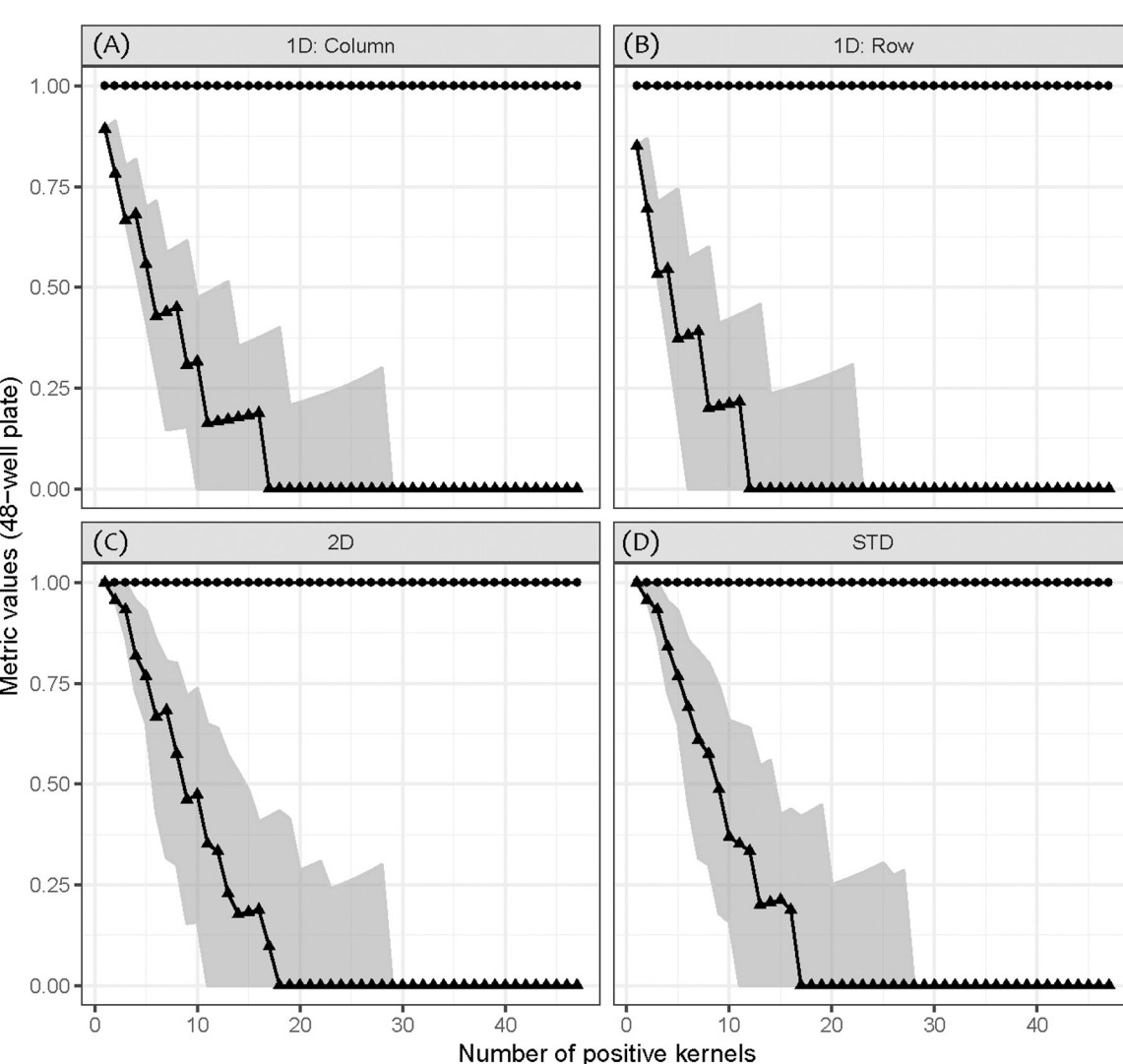

**Fig 2. Sensitivity and specificity of four pooling strategies at different prevalence levels for 48-well plates.** The point represents the median and the shaded area represents the range from 2.5th percentile to 97.5th percentile. (A) 1D pooling by columns. (B) 1D pooling by rows. (C) 2D pooling. (D) STD-pooling (n = 48; q = 7; k = 2).

than without pooling while 2D and STD pooling would need 2.3–3.3 folds more pipetting. In terms of 96-well plates, the pattern of fold change in 1D and 2D pooling was similar to what was observed in the 48-well plates. However, STD pooling would demand much more pipetting, ranging from 3.3–4.3 folds (S6 Fig).

## Simulated pooling was validated with experimental pooling

Sensitivity and specificity values calculated from the experimental pooling data were compared with those from the simulation to determine whether the simulation could make accurate estimation (Table 4).

For 1D and 2D pooling validation, the experimental sensitivities were 100% for all three pooling strategies, which were the same as the medians of the simulated sensitivities. While the

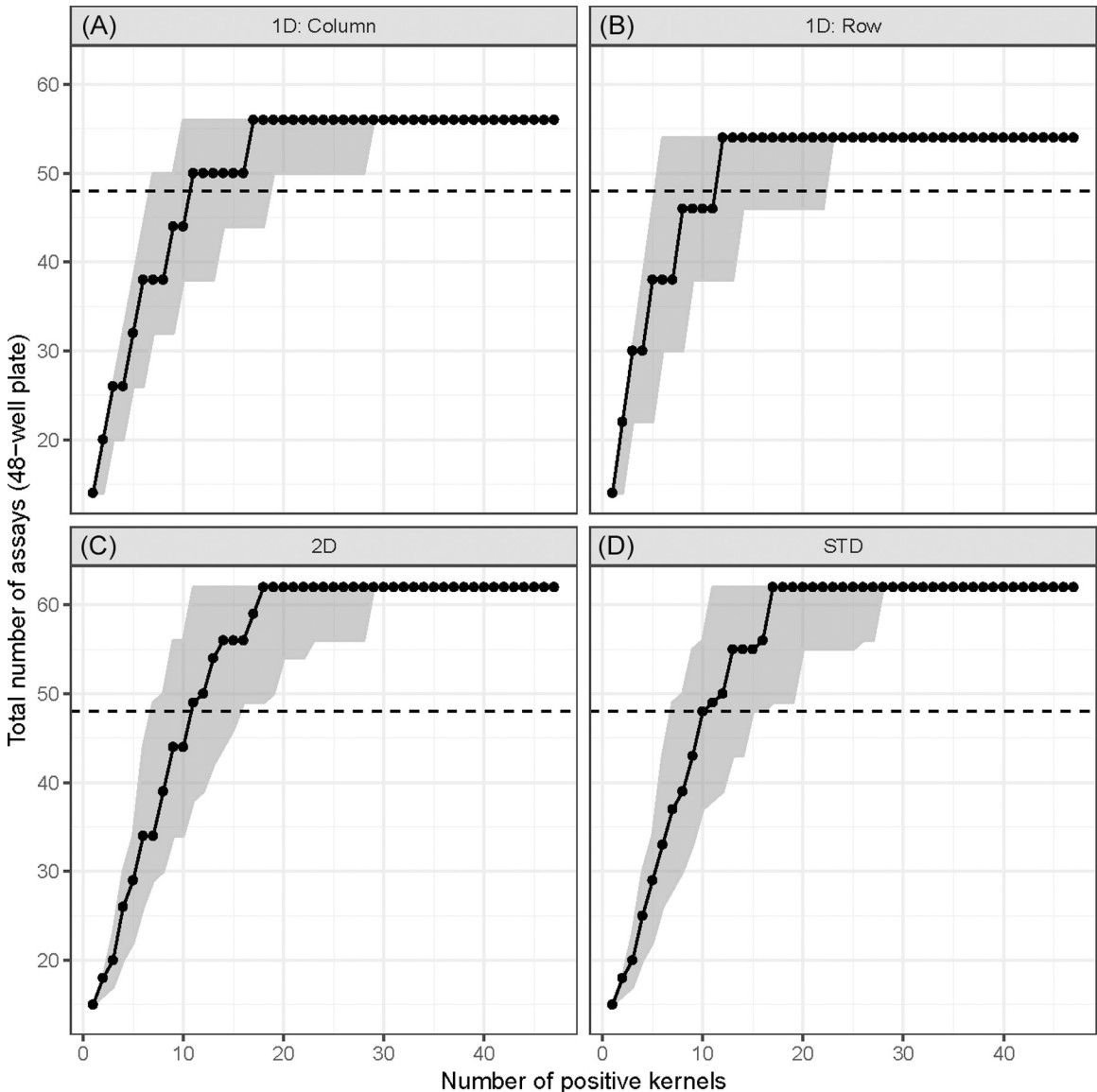

**Fig 3. Total number of assays needed for four pooling strategies at different prevalence levels for 48-well plates.** The dashed line indicates that 48 assays would be needed without pooling. The dot represents the median and the shaded area represents the range from 2.5th percentile to 97.5th percentile. (A) 1D pooling by columns. (B) 1D pooling by rows. (C) 2D pooling. (D) STD-pooling (n = 48; q = 7; k = 2).

specificities of 1D pooling by column (39%) and 2D pooling (39%) were higher than the median of the simulated specificity (0%), all the experimental specificities still fell within the acceptable range.

In terms of validating STD pooling, only 6 out of 48 kernels are above 20 ppb and are identified as aflatoxin-positive kernels. This prevalence level suggested STD (n = 48; q = 7; k = 4) was a suitable pooling scheme. Again, the experimental sensitivity was the same as the median of the simulated one (100%). While the experimental specificity (86%) was lower than the median of the simulated specificity (95%), it was still within the acceptable range.

Overall, the validation of simulated metrics against the experimental ones showed that the simulated sensitivities were highly accurate and precise in predicting experimental sensitivity

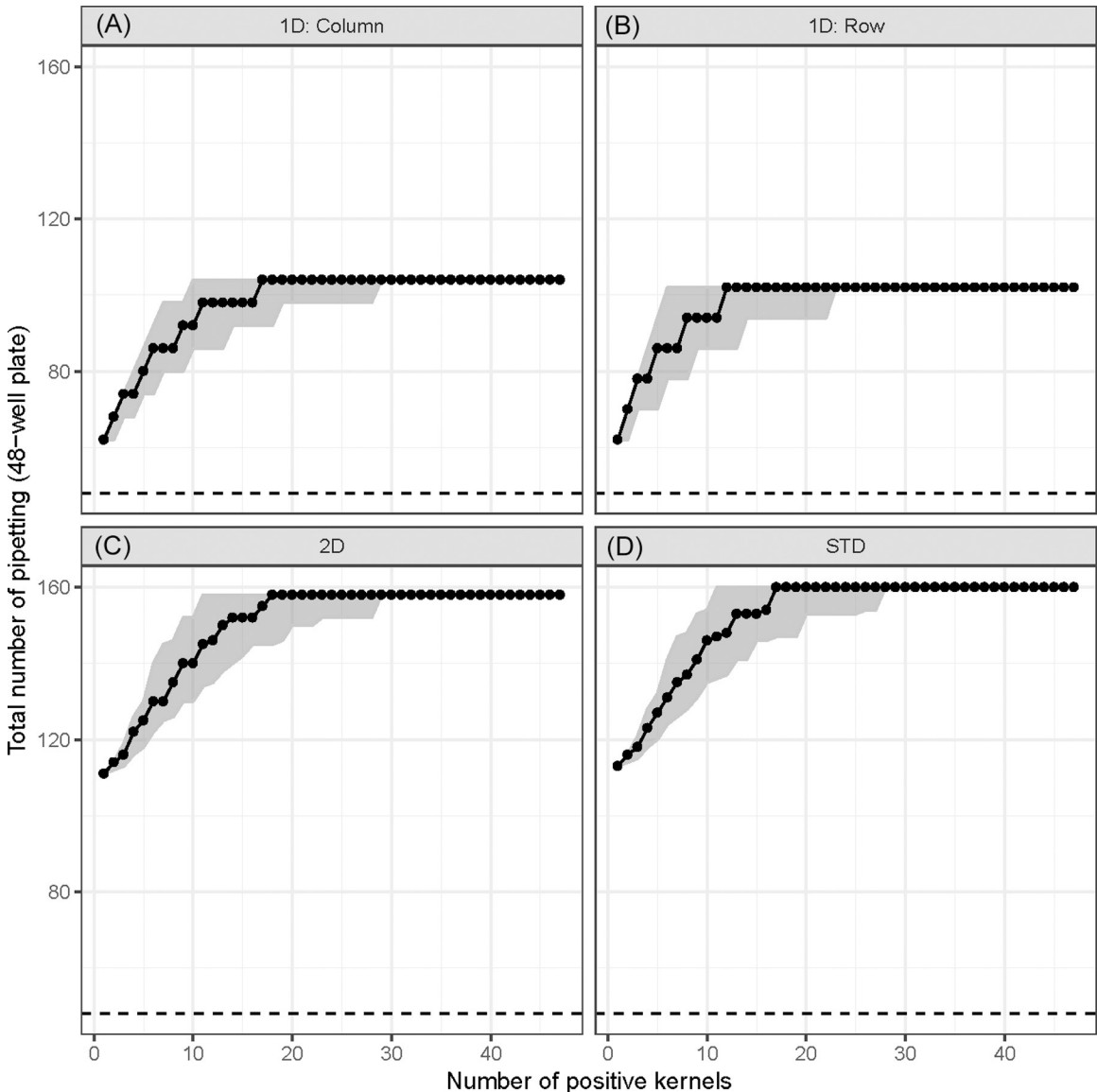

**Fig 4. Total number of pipetting for four pooling strategies at different prevalence levels for 48-well plates.** The dashed line indicates that without pooling 48 times of pipetting would be needed to transfer samples into the assay plate. The dot represents the median and the shaded area represents the range from 2.5[th] percentile to 97.5[th] percentile. (A) 1D pooling by columns. (B) 1D pooling by rows. (C) 2D pooling. (D) STD-pooling ($n = 48$; $q = 7$; $k = 2$).

whereas the simulated specificities may be less precise in prediction, albeit within the allowable range.

## Discussion

### All pooling strategies can detect positive samples without false negatives

1D, 2D, and STD ($n$; $q$; $k$ = 48; 7; 2 and 96; 5; 3) pooling can detect all contaminated kernels (> 20 ppb) without false negatives. This finding relieves experimenters of the concern that pooling would dilute the positive samples and result in failure of detection. The finding would remain valid regardless of the actual prevalence level, which is usually unknown prior to

**Table 4. Validation of simulated sensitivity and specificity against empirical metrics for four types of pooling strategies in 48-well plates.**

| Pooling strategy | Mycotoxin | Number of positive kernels[a] | Pooling sensitivity | | Pooling specificity | |
|---|---|---|---|---|---|---|
| | | | Experiment | Simulation (Median with inner fences)[b] | Experiment | Simulation (Median with inner fences)[b] |
| 1D Row | Fumonisin | 17 (35%) | 100% | 100% (100% - 100%) | 0% | 0% (0% - 0%) |
| 1D Column | Fumonisin | 17 (35%) | 100% | 100% (100% - 100%) | 39% | 0% (0% - 48%) |
| 2D | Fumonisin | 17 (35%) | 100% | 100% (100% - 100%) | 39% | 0% (0% - 48%) |
| STD (48; 7; 4) | Aflatoxin | 6 (13%) | 100% | 100% (100% - 100%) | 86% | 95% (80% - 100%) |

[a] Prevalence of positives is presented in the parentheses.

[b] The inner fences include the lower inner fence (Q1–1.5 × IQR) and the upper inner fence (Q3 + 1.5 × IQR), where Q1 = 25[th] percentile, Q3 = 75[th] percentile, and IQR = Q3 –Q1. The simulation is considered valid when the experimental metrics fall within the inner fences of the simulated metrics.

experiments. However, it is worth mentioning that there is a limit to the pooling size in the context of measuring aflatoxin with ELISA assays. The conclusion would only hold when a pool satisfies the following inequality: $\frac{positive\ individual\ threshold}{pooling\ size} \geq LOD$, where LOD is the limit of detection of the assay. This paper used 20 ppb as the positive aflatoxin threshold to comply with the U.S. regulations for corn intended for human consumption and 1 ppb as the LOD. These values yield a maximum pooling size of 20. For countries or regions with more stringent regulation on aflatoxin, the maximum pooling size will be smaller. For example, the EU's aflatoxin regulatory limit is 4 ppb for groundnuts intended for direct human consumption and thus the maximum pooling size for ELISA will be 4 [20]. For other assays, there may be different criteria for choosing a proper pooling size, such as using $IC_{50}$ values for enzyme screening [21].

Our study only considers scenarios where the pooling size is within the allowable limit. In our simulation, the conclusion of 100% sensitivity remains valid even when $\frac{positive\ individual\ threshold}{pooling\ size}$ is at the LOD. This may not be true in real experiments and one could observe false negatives, given that the positive signal is blurry when the analyte concentration is near the LOD. When $\frac{positive\ individual\ threshold}{pooling\ size}$ is smaller than LOD, it is highly likely to have false negatives. Researchers using the guidance would need to determine pooling size limit for their specific assay and select an appropriate pooling strategy.

This study explores the advantages of pooling on the foundation of single kernel mycotoxin measurement, which is a method adopted by several studies investigating the plausibility of high-throughput mycotoxin detection [10,22–24]. Admittedly, it is more common in the industry to use representative samples rather than single kernels to estimate the overall mycotoxin level in bulk corn. Mycotoxin level in a representative sample tends to be less extreme due to the fact that it is formed by compositing multiple small portions of grain. Nevertheless, mycotoxin level in bulk corn has been reported to be skewed; while many composite samples are low in mycotoxin, a few samples may exceed the regulatory limit and reach an astonishingly high level [25,26]. This creates an opportunity for pooling as it equates to a scenario with low-prevalence events, which fits the precondition for pooling. Theoretically, the conclusion of 100% sensitivity and decreasing specificity should still hold true for pooling representative bulk samples as long as the aforementioned inequality is satisfied.

## Trade-off between cost of reagent and cost of pipetting

It has been reported in multiple studies that pooling can lower the cost of reagents to varying degrees in large-scale experiments [4,5,27]. In this study, pooling would reduce the number of

assays by around 69% - 71% for a 48-well plate and around 78% - 83% for a 96-well plate at most. This could potentially alleviate the economic burden of testing large quantities of samples with low prevalence in a resource-limited setting. However, the cost saving effect would diminish when the prevalence increases because there would be more positive pools and thus more re-tests to perform. Such a finding corroborates a previous study's result that lower prevalence leads to higher proportion of tests saved at a fixed pool size [6]. Beyond the critical prevalence level, pooling would backfire and cost more in reagents than without pooling. Hence, prior knowledge of prevalence level, often acquired from empirical data, would be useful to determine whether pooling is likely beneficial.

Potential savings in reagents come with a price of increasing cost in pipetting. These four pooling strategies would multiply the number of pipetting by 1.3–3.3 folds for a 48-well plate and 1.2–4.3 folds for a 96-well plate. Compared to 1D pooing, 2D and STD pooling tend to perform slightly better at reducing reagent cost, but they would also require much more pipetting. This may defeat the purpose of cost-saving if the cost incurred by extra labor work or extended working hours becomes substantial, let alone the empirical fact that long hours of repetitive pipetting is an error-prone process.

To reconcile the conflict between cost of reagent and cost of pipetting, it may be helpful to estimate the total cost of pooling and make comparison with the cost of non-pooling. This study, along with previous research, suggests that prevalence level in the sample has a heavy influence on test specificity, which drives the cost of reagent and pipetting [6]. With the estimated prevalence level, one can find the corresponding median number of assays and pipetting from the simulation results. These estimates could be used to calculate the expected total cost of pooling, which may be expressed as a weighted sum incorporating the full cost of reagents, cost of pipetting, and other miscellaneous costs including but not limited to human labor, experiment time, equipment purchasing, and maintenance. Depending on the experimental setting, each laboratory may put unique weights on these sources of expenditure and thus the same experimental task could result in different total costs. It is only advisable to implement pooling when the expected total cost of pooling is lower than the total cost of without pooling.

## Automation

In cases where pooling is required for multiple batches of sample, automating the pooling process may be a good investment. It would not only remarkably reduce the workload for lab technicians but also perform pooling with increased precision and lower risk of pipetting error that human experimenters could achieve. Applications of automated pipetting have been well developed and utilized in large-scale experiments, such as high throughput screening for target molecules in the pharmaceutical industry. Robotic platforms with integrated chemical assay workstation have been built and used to facilitate chemical and biological profiling of potential drug compounds [2]. A more recent study of utilizing pipetting robots to perform assay on antioxidants has also shown great possibility that these automated systems can be adapted to conduct pooling in a medium-scale experiment setting [28].

## Conclusions

Pooling is widely adopted in large-scale chemical experiments with the purpose of rapid screening and cost reduction [19]. While pooling theories have been well established and validated in a large-scale experimental setting, there remains an opportunity for practical instruction of pooling in medium-scale experiments [8]. Our study strives to provide detailed instruction for pooling in a medium scale setting and examines how different pooling strategies affect assay performance, reagent cost, and cost of pipetting.

In our Monte Carlo simulation, we simulated aflatoxin-contaminated kernels from fitted distributions and used four different pooling strategies to pool samples before subjecting them to aflatoxin detection assay. Simulation results demonstrate that all positive samples can be detected after pooling by all four strategies (1D by rows, 1D by columns, 2D, STD), regardless of the actual prevalence level. Prevalence level affects false negative rate and plays a major role in deciding whether pooling would save reagents. Pooling could reduce the cost of reagent by around 70% in 48-well plates and 80% in 96-well plates. When there is only one positive sample, 1D and STD pooling would be the most cost-efficient for 48- and 96-well plates respectively. However, such a cost-saving effect would be achieved only when prevalence level is lower than approximately 21% for experiments conducted in 48-well plates and a range of 13% - 21% for 96-well plates, with STD pooling being most sensitive to prevalence level and 1D pooling by column most tolerant. Meanwhile, pooling would inevitably increase the number of pipetting by 1.3–3.3 folds for 48-well plates and 1.2–4.3 folds for 96-well plates. More specifically, the inflation of pipetting cost is the mildest when 1D pooling by row is implemented while STD pooling is likely to cause a surge in pipetting cost. Because prevalence level has a substantial influence on reagent and pipetting cost, it is crucial to estimate the prevalence level of sample so that the experimenter can calculate the total cost of pooling. With the total cost, the experimenter may decide whether pooling is appropriate and which pooling strategy to employ if pooling is worth trying.

Generally, 1D pooling would be the most cost-saving for both 48- and 96-well plates when considering the reagent-pipetting trade-off. STD pooling, however, could be a better alternative when automation is available to offset the soaring cost incurred by extra pipetting.

## Supporting information

**S1 Fig. The STD pooling scheme for 48 samples with 1 expected positive sample, 0 expected error, and a maximum of 8 extracts allowed to be pooled.** The horizontal dashed lines split the pooling scheme into 2 layers. There are 14 pools; each layer contains 7 pools and each pool comprises a combination of samples indicated as squares.
(TIF)

**S2 Fig. Comparison between the distribution of real aflatoxin data and simulated data.** The histogram represented the real aflatoxin concentration distribution, with 6 kernels (4%) $\geq$ 20 ppb aflatoxin and 132 kernels (96%) < 20 ppb aflatoxin. The density plot (grey shaded area) illustrated the distribution of simulated data with $9.6 \times 10^5$ healthy kernels (96%) and $4 \times 10^4$ contaminated kernels (4%).
(TIF)

**S3 Fig. Comparison between the experimental and simulated distribution of fumonisin.** The histogram represented the experimental fumonisin concentration with 43 kernels (46%) $\geq$ 1 ppm and 50 kernels (54%) < 1 ppm. The density plot (grey shaded area) illustrated the distribution of simulated data with $5.4 \times 10^5$ (54%) healthy kernels and $4.6 \times 10^5$ (46%) contaminated kernels.
(TIF)

**S4 Fig. Sensitivity (circle) and specificity (triangle) at different levels of prevalence for 96-well plates.** The point represents the median and the shaded area represents the range from 2.5[th] percentile to 97.5[th] percentile. Top left panel is 1D pooling where columns are pooled, top right panel is 1D pooling where rows are pooled, bottom left panel is 2D pooling, and bottom right panel is STD-pooling (n = 96; q = 5; k = 3).
(TIF)

**S5 Fig. Total number of assays needed at different levels of prevalence for 96-well plates.**
The dashed line indicates that 96 tests would be needed without pooling. The dot represents
the median and the shaded area represents the range from 2.5th percentile to 97.5th percentile.
Top left panel is 1D pooling where columns are pooled, top right panel is 1D pooling where
rows are pooled, bottom left panel is 2D pooling, and bottom right panel is STD-pooling
($n = 96$; $q = 5$; $k = 3$).
(TIF)

**S6 Fig. Total number of pipettings at different levels of prevalence for 96-well plates.** The
dashed line indicates that without pooling 96 times of pipetting would be needed to transfer
samples into ELISA assay plate. The dot represents the median and the shaded area represents
the range from 2.5th percentile to 97.5th percentile. Top left panel is 1D pooling where columns
are pooled, top right panel is 1D pooling where rows are pooled, bottom left panel is 2D pool-
ing, and bottom right panel is STD-pooling ($n = 96$; $q = 5$; $k = 3$).
(TIF)

**S1 Table. The STD-pooling scheme ($n = 48$; $q = 7$; $k = 2$) in table format for 48 samples.**
Each pool consists of 7 samples of equal volume. Number 1 to 48 represents the sample index
and number 0 represents the solvent (80% methanol solution).
(DOCX)

**S2 Table. Glossary of important Shifted Transversal Design parameters.**
(DOCX)

**S1 Appendix.**
(DOCX)

## Acknowledgments

We thank Tim Herrman from the Office of the Texas State Chemist at Texas A&M AgriLife
Research for providing us with commercial corn from Texas counties.

## Author Contributions

**Conceptualization:** Xianbin Cheng, Matthew J. Stasiewicz.

**Data curation:** Xianbin Cheng, Ruben A. Chavez.

**Formal analysis:** Xianbin Cheng.

**Funding acquisition:** Matthew J. Stasiewicz.

**Investigation:** Xianbin Cheng.

**Methodology:** Xianbin Cheng.

**Project administration:** Matthew J. Stasiewicz.

**Resources:** Matthew J. Stasiewicz.

**Software:** Xianbin Cheng.

**Supervision:** Matthew J. Stasiewicz.

**Validation:** Xianbin Cheng, Ruben A. Chavez.

**Visualization:** Xianbin Cheng.

**Writing – original draft:** Xianbin Cheng.

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
