## [Decision Letter · Decision Letter 0]

18 May 2020

PONE-D-20-06523

When to use one-dimensional, two-dimensional, and Shifted Transversal Design pooling in mycotoxin detection

PLOS ONE

Dear Dr Stasiewicz,

Thank you for submitting your manuscript to PLOS ONE. After careful consideration, we feel that it has merit but does not fully meet PLOS ONE’s publication criteria as it currently stands. Therefore, we invite you to submit a revised version of the manuscript that addresses the points raised during the review process.

ACADEMIC EDITOR: 

Unfortunately, we are unable to publish the manuscript in its current form as both reviewers raised substantial technical and methodological concerns. We would be willing to consider a revised manuscript that addresses all the issues raised by both reviewers. Addressing Reviewer 1's comments about the appropriateness of the pooling strategy is of critical importance.

We would appreciate receiving your revised manuscript by Jul 02 2020 11:59PM. To enhance the reproducibility of your results, we recommend that if applicable you deposit your laboratory protocols in protocols.io, where a protocol can be assigned its own identifier (DOI) such that it can be cited independently in the future. For instructions see: http://journals.plos.org/plosone/s/submission-guidelines#loc-laboratory-protocols

We look forward to receiving your revised manuscript.

Kind regards,

Jishnu Das, Ph.D.

Academic Editor

PLOS ONE

Journal Requirements:

Reviewers' comments:

Reviewer's Responses to Questions

**Comments to the Author**

1. Is the manuscript technically sound, and do the data support the conclusions?

Reviewer #1: No

Reviewer #2: Partly

2. Has the statistical analysis been performed appropriately and rigorously? 

Reviewer #1: I Don't Know

Reviewer #2: I Don't Know

3. Have the authors made all data underlying the findings in their manuscript fully available?

Reviewer #1: Yes

Reviewer #2: Yes

4. Is the manuscript presented in an intelligible fashion and written in standard English?

Reviewer #1: Yes

Reviewer #2: Yes

5. Review Comments to the Author

Reviewer #1: This manuscript attempts to address the question of whether and in what conditions pooling strategies are beneficial in the context of medium-scale experiments, taking into account the costs of pipetting from building the pools in addition to the costs of the assays themselves. This is a very interesting practical question. Unfortunately the manuscript fails to provide useful answers due to methodological flaws.

The main issue is that the pools used are inappropriate for the conditions considered in the manuscript. Indeed, when performing experiments on pools, the pool size should obviously depend on the expected number of positive samples, ie on the prevalence. For example with a prevalence of 35% as in Table 4 for fumonisin, any pool of 8 samples has 1 - 0.65^8 = 97% chance to contain a positive sample. Therefore building and testing such pools (eg 1D Row in Table 4) hardly provide any information at all: virtually every pool will be positive, hence each sample will need to be individually re-tested. There is no need for any kind of simulation or experiment to establish this.

Similarly, in this study the pool size of STD designs is set once and for all assuming a single contaminated sample (as explained P6), corresponding to a prevalence of 2% in 48-well format and 1% in 96-well format. The authors then use these pools in high prevalence situations, which obviously cannot work for the same reasons outlined above: every pool will be positive and nothing is learned from the pooling results.

Although the exact number of positives is not known, one can usually have a rough estimate, and this is sufficient to design informative pools. For 1D and 2D designs the experimenter has limited control and can only choose the assay format (eg 48 or 96). However with STD designs the pool size can be set as desired by choosing the value of q. Given the expected prevalence, the standard procedure is to choose q such that approximately 50% of pools will be positive, while respecting the maximum pool size due to the assay LOD. For example, when testing aflatoxin contamination (expecting ~13% prevalence) one should aim for pools of size 5, therefore in 96 format a reasonable design could be STD(96;19;2). Testing such a design could provide interesting results, contrary to STD(96;5;3).

Other less critical issues follow.

- The manuscript is somewhat confusing between simulations and experiments. Could the authors please clarify what was studied through simulation and what exactly was tested experimentally. A figure representing the study design would be helpful for this.

- Concerning STD pools with n=48, the manuscript mostly uses STD(48;7;2) (eg Table 3) but in some places (eg Table 4) it uses STD(48;7;4) instead. Please clarify.

- Table 3 presents formulae for the number of pipetting, followed by Min and Max columns which should match, but they do not. For example the 1D Column and 1D Row formulas are identical, this must be wrong. Also the formulae for STD do not match the Min/Max values at all.

- The manuscript describes a method for interpreting the STD pooling results based on 2 criteria (P7). This method marks samples as positive or negative, and also leaves some samples unmarked. When every pool is positive this will result in every sample remaining unmarked, therefore there should be zero false positives when prevalence is high (contrary to the presented results). I suspect the authors apply a third criteria, where unmarked samples are considered positive? If so, it needs to be explained and justified.

Reviewer #2: The topic of the paper is original and gives a theoretical and experimental study on different sample pooling techniques for mycotoxin screening through ELISA. It is fluently written. Nevertheless I have major questions and comments which I would like to be clarified.

First of all, a change in the title is needed: instead of detection, ‘screening’ is preferred, as this type of pooling only works for a screening technique with a dedicated cut-off level.

The abstract should make clear from the beginning that the focus of the work is on ELISA (this is only mentioned in the middle of the abstract). It is also better to write one-dimensional (1D) and two-dimensional (2D) in line 18.

GENERAL MAJOR COMMENTS

A major comment that I have on the paper is that a generalization is made that positive samples in mycotoxin analysis are rare (eg in the Introduction line 44). This however depends on the legislative maximum limits used as well on the region. Indeed, in line 67 you write that aflatoxin contamination of corn kernels is a low-prevalence event. I cannot agree that this generalization is made; eg in some regions in Africa this is a very high-prevalence event. This should be more diversified in the paper.

Another major comment is on the 20 ppb standard which is set by the FDA. As you know in other parts of the world, such as the European Union, maximum limits for aflatoxins are much lower; in that event, the pooling concept will probably not work, as the detection limit of ELISA might not be sensitive enough. It will be needed to discuss this in the paper, so that it can be interesting for broader set of readers.

Another major comment concerns that the study is based on the analysis of single kernels. This is not how routine control for mycotoxins in corn is generally performed. In order to have representative samples, corn lots are ground and different subsamples are analysed. So, I am really not convinced that the presented pooling can work in that case? As you work with representative samples, you won’t have those extremely high mycotoxin concentrations as you had in the study in one single kernel (as mentioned on p17, line 337: 4.0 x 104 ppb for aflatoxin B1 and 36 ppm for fumonisins). Please also have this in the discussion section.

Another major comment is on the contamination of the single corn kernels; on line 439 you write “we contaminated single corn kernels with different level of aflatoxin”. So, these were not naturally contaminated? And what about the fumonisins? Of course, if you work with spiked samples, this will also have an influence on the extraction, and will perhaps lead to much better reproducible results than with naturally contaminated ones. If you spiked yourself, you need to mention the method how this was done. On lines 139-141 you mention low-aflatoxin class, medium and high; where I had the impression that these were naturally contaminated; so a lot of confusion which needs to be cleared.

SPECIFIC COMMENTS

In the Materials and Methods section, it is needed to mention the Helica ELISA test which is used, more in the beginning of the description, so that it is clear that an ELISA with 1 ppb LOD has been used for the whole study.

I am also wondering that on p7, line 125, for the 96-well the positive-pool threshold is set at the LOD. So, what about false results when working at the LOD?

In Line 135, suddenly also fumonisins are mentioned, while before only aflatoxins were mentioned; better to make clear from the start that both mycotoxin groups are part of the study.

Line 206: what is this ‘maximum limit of ELISA’?

I was wondering why the 1D and 2D pooling strategies were only validated with fumonisins and the STD only with aflatoxin?

Line 261: why only >20 ppb aflatoxin is mentioned, and nothing about fumonisin?

6. PLOS authors have the option to publish the peer review history of their article (what does this mean?). If published, this will include your full peer review and any attached files.

Reviewer #1: No

Reviewer #2: No

---

## [Author Response · Author response to Decision Letter 0]

22 Jun 2020

Response uploaded as attachement

---

## [Decision Letter · Decision Letter 1]

13 Jul 2020

When to use one-dimensional, two-dimensional, and Shifted Transversal Design pooling in mycotoxin screening

PONE-D-20-06523R1

Dear Dr. Stasiewicz,

We’re pleased to inform you that your manuscript has been judged scientifically suitable for publication and will be formally accepted for publication once it meets all outstanding technical requirements.

Kind regards,

Jishnu Das, Ph.D.

Academic Editor

PLOS ONE

Additional Editor Comments (optional):

Reviewers' comments:

Reviewer's Responses to Questions

**Comments to the Author**

1. If the authors have adequately addressed your comments raised in a previous round of review and you feel that this manuscript is now acceptable for publication, you may indicate that here to bypass the “Comments to the Author” section, enter your conflict of interest statement in the “Confidential to Editor” section, and submit your "Accept" recommendation.

Reviewer #1: All comments have been addressed

Reviewer #2: All comments have been addressed

2. Is the manuscript technically sound, and do the data support the conclusions?

Reviewer #1: (No Response)

Reviewer #2: (No Response)

3. Has the statistical analysis been performed appropriately and rigorously? 

Reviewer #1: (No Response)

Reviewer #2: (No Response)

4. Have the authors made all data underlying the findings in their manuscript fully available?

Reviewer #1: (No Response)

Reviewer #2: (No Response)

5. Is the manuscript presented in an intelligible fashion and written in standard English?

Reviewer #1: (No Response)

Reviewer #2: (No Response)

6. Review Comments to the Author

Reviewer #1: (No Response)

Reviewer #2: (No Response)

7. PLOS authors have the option to publish the peer review history of their article (what does this mean?). If published, this will include your full peer review and any attached files.

Reviewer #1: No

Reviewer #2: No

---

## [Editor Report · Acceptance letter]

23 Jul 2020

PONE-D-20-06523R1 

When to use one-dimensional, two-dimensional, and Shifted Transversal Design pooling in mycotoxin screening 

Dear Dr. Stasiewicz:

I'm pleased to inform you that your manuscript has been deemed suitable for publication in PLOS ONE. Congratulations! Your manuscript is now with our production department. 

Kind regards, 

on behalf of

Dr. Jishnu Das 

Academic Editor

PLOS ONE